# Comparative Sample Preparation Using Focused Ion Beam and Ultramicrotomy of Human Dental Enamel and Dentine for Multimicroscopic Imaging at Micro- and Nanoscale

**DOI:** 10.3390/ma15093084

**Published:** 2022-04-24

**Authors:** Katharina Witzke, Marcus Frank, Olaf Specht, Ute Schulz, Claudia Oehlschläger, Detlef Behrend, Peter Ottl, Mareike Warkentin

**Affiliations:** 1Department of Oral and Maxillofacial Surgery/Plastic Surgery, Greifswald University Medicine, Ferdinand-Sauerbruch-Straße DZ7, 17475 Greifswald, Germany; katharina.witzke@med.uni-greifswald.de; 2Electron Microscopy Centre, Rostock University Medical Center, Strempelstraße 14, 18057 Rostock, Germany; marcus.frank@med.uni-rostock.de (M.F.); ute.schulz@med.uni-rostock.de (U.S.); 3Department of Life, Light and Matter, University of Rostock, Albert-Einstein-Straße 25, 18059 Rostock, Germany; detlef.behrend@uni-rostock.de (D.B.); peter.ottl@med.uni-rostock.de (P.O.); 4Department of Material Science and Medical Engineering, University of Rostock, Friedrich-Barnewitz-Straße 4, 18119 Rostock, Germany; specht@fbn-dummerstorf.de (O.S.); claudia.oehlschlaeger@uni-rostock.de (C.O.); 5Institute of Behavioural Physiology, Leibnitz Institute for Farm Animal Biology (FBN), Wilhelm-Stahl-Allee 2, 18196 Dummerstorf, Germany; 6Department of Prosthodontics and Materials Sciences, Rostock University Medical Center, Strempelstraße 13, 18057 Rostock, Germany

**Keywords:** enamel, dentine, focused ion beam, ultramicrotomy, TEM, artefacts

## Abstract

(1) Background: The aim of this study was to systematically compare TEM sections of mineralized human enamel and dentine prepared by focused ion beam (in situ lift-out) technique and ultramicrotomy through a combination of microscopic examination methods (scanning electron microscopy and transmission electron microscopy). In contrast with published studies, we compared the TEM preparation methods using the same specimen blocks as those for the ultramicrotomy and FIB technique. (2) Methods: A further evaluation of TEM sample preparation was obtained by confocal laser scanning microscopy and atomic force microscopy. In addition, ultramicrotome- and focused ion beam-induced artefacts are illustrated. (3) Results: The FIB technique exposed a major difference between non-decalcified enamel and dentine concerning the ultrastructural morphology compared to ultramicrotome-prepared sections. We found that ultramicrotomy was useful for cutting mineralized dentine, with the possibility of mechanical artefacts, but offers limited options for the preparation of mineralized enamel. FIB preparation produced high-quality TEM sections, showing the anisotropic ultrastructural morphology in detail, with minor structural artefacts. Our results show that the solution of artificial saliva and glutardialdehyde (2.5% by volume) is a very suitable fixative for human mineralized tissue. (4) Conclusions: The protocol that we developed has strong potential for the preparation of mineralized biomaterials for TEM imaging and analysis.

## 1. Introduction

The main concern in dental biomaterial research at present is the development of materials and biomaterials that present appropriate ultrastructural, mechanical, physicochemical and biological characteristics. Dental materials with mechanical and structural compatibility, along with their corresponding dental hard tissues (enamel and dentine), present a wealth of potential in terms of their clinical applications in the field of prosthodontics and conservative dentistry. These materials must meet the requirements of the target tissue, with the aim of significantly improving the longevity and functionality of the restoration. The implementation of dental materials with optimal properties includes precise knowledge and structural analysis of the morphology of dental hard tissues from the macroscale down to the nanoscale, which is very challenging, particularly without prior demineralization. Human mature enamel is a non-vital, hydrated (3% by weight), highly filled biological composite consisting of inorganic hydroxyapatite crystallites (96% by weight) embedded in an organic matrix containing enamel-specific proteins (1% by weight), as in [1,2,3]. This is the human body’s most highly mineralized hard tissue and is difficult to mechanically cut to transmit electron microscopy (TEM) using an ultramicrotome. In terms of structural hierarchy, human mature enamel consists of several levels (Figure 1a–d) [4].

At the nanoscale, human mature enamel is composed of hexagonal hydroxyapatite crystallites (level 1, Figure 1d), which are 26.3 ± 2.2 nm thick and 68.3 ± 13.4 nm wide [5]. They are approximately arranged parallel to each other into nanofibrils, with a length between 0.1 and 1 µm and typical diameters of 30–40 nm (level 2), 80–130 nm (level 3) and 800 nm (level 4). At the microscale, the nanofibrils are arranged in an alternating orientation, forming prisms and interprisms (level 5, Figure 1b,c) [4]. Prism rods (level 6) are arranged perpendicular to the tooth surface, with an increasing twist to compensate for the volume difference between the dentine–enamel junction (DEJ) and the outer enamel surface [6]. This alternating direction of the prism rods reveals the Hunter–Schreger bands (level 7, Figure 1b) [4].

However, human mature dentine is a hydrated (10% by weight), vital and middle-filled biocomposite (Figure 1a,e–g) consisting of inorganic hydroxyapatite crystallites (70% by weight) embedded in an organic matrix (20% by weight) containing collagen-type 1 (90%) and non-collagenous proteins (10%) that are visible on the nanoscale (Figure 1g) [3]. Based on microscopic observations, the microstructure of dentine reveals numerous dentinal tubules (Figure 1e,f) with a diameter of 1–2 µm containing the odontoblast processes, which extend to the inner third crown dentine [7]. A microscopic examination revealed a structural hierarchy of human enamel and dentine, containing anatomical structures from the macroscale to the nanoscale (Figure 1). A detailed knowledge of these mechanical properties and of the native ultrastructural morphology of enamel and dentine is essential to further improve the development of dental restorative materials at the corresponding mechanical and structural levels. The quasi-statistic and dynamic mechanical parameters of human dental hard tissues were documented in earlier reports [8,9,10]. Transmission electron microscopy (TEM) is a commonly used method for ultrastructural examinations at the nanoscale. In the literature, various efforts have been made to prepare ultrathin human enamel and dentine sections for TEM imaging [11,12,13,14,15,16,17,18,19,20,21,22,23]. Most of these studies used different specimen-preparation protocols concerning the nature of the hard tissue (origin), storage medium after collection, fixation, rinsing, dehydration, embedding and target preparation. Since the 1960s, diamond knives have been employed for the preparation of ultrathin sections with ultramicrotomy and have helped to examine the ultrastructure of mineralized and demineralized human mature enamel and dentine with TEM [13,14,15,24]. Diamond knives can introduce artefacts to the specimen structure, for example, specific chatter marks, due to the high degree of mineralization and hardness in the sample [22]. However, specimen preparation using ultramicrotomy can be improved by partly demineralizing tissues, such as bone, enamel, dentine and cementum, with ethylenediaminetetraacetic acid (EDTA) [7,17].

Focused ion beam (FIB) milling is an alternative specimen-preparation method, as opposed to conventional preparation using ultramicrotomy. This technique allows for the high-precision preparation of a TEM lamella (approximately 15 × 10 × 0.15 µm) by specimen removal at nanometer steps. The FIB instrument is equipped with a scanning electron microscope for in situ control of the workflow and specimen position. The FIB technique was successfully used to produce and investigate ultrathin mineralized human enamel sections using TEM, as reported by HAYASHI et al. [19], followed by HOSHI et al.’s [20] preparation of mineralized human dentine. Studies comparing the enamel and dentine sections prepared by ultramicrotomy and the focused ion beam technique are rare. Recent comparative studies include ivory dentine and dehydrated maxillary/mandibular premolars using TEM [22,25], as well as energy-dispersive spectroscopy of X-ray (EDS) and electron energy-loss spectroscopy (EELS) [22] for an analytical evaluation of the preparation results. The aim of this study was to develop a TEM-preparation protocol, obtaining as precise native structure information as possible. In the present study, we systematically compared ultramicrotomy and FIB preparation from the micro- to nanoscale using a combination of various microscopic examination methods, including AFM, CLSM, SEM and TEM. Finally, preparation-induced artefacts and their consequences for TEM examination were analyzed. The originality of this study lies in the fact that we compared TEM preparation methods using the same specimen blocks for ultramicrotomy and the in situ FIB lift-out technique.

## 2. Materials and Methods

### 2.1. Preparation of Human Dental Enamel and Dentine

Mandibular premolars (FDI 34, 44, 14 and 24) of a 15-year-old patient exhibiting no systemic diseases were extracted for orthodontic reasons. The teeth were fully erupted without any carious lesions or fillings. This study was approved by the ethics committee of the University Medicine Rostock (reference No A 2010 26). To prevent drying of artefacts, the premolar was stored in artificial saliva (DAC/NRF 7.5) at 4 °C immediately after extraction.

Microscopic documentation was performed prior to preparation of the dental tissue using a stereo zoom microscope (Stereozoom SZX 10, light source: KL1500LCD; Olympus, Hamburg, Deutschland) and microcomputed tomography (SkyScan 1172, software version 1.5; Bruker, Billerica, MA, USA) with the following parameters: 80 kV, 100 µA, filter Al 0.5 mm, rotation 360° and an image voxel size of about 3 × 34.5 µm to ensure an intact enamel and dentine structure (Figure 2). The tooth was fixed in a solution of artificial saliva and glutardialdehyde (2.5 vol%) for 14 days at 4 °C and then rinsed with Soerensen’s buffer for 7 days at 4 °C with a renewing interval of 48 h. For the preparation of dental tissue, the premolar was embedded in a two-component, cold-curing resin system (Epo Thin^®^ 2 resin and hardener; Buehler, Düsseldorf, Germany) for further mechanical conditioning. To avoid infiltration of the epoxy resin and drying of artefacts during polymerization (24 h at 20 °C), the specimens were moistened with artificial saliva and wrapped in Parafilm (Parafilm “M“^®^; Pechiney Plastic Packaging, Chicago, IL, USA) before the embedding process. Using a low-speed diamond wire saw (Precision Diamond Wire Saw 3241; Well, Mannheim, Deutschland), the cured blocks were cut longitudinally to the center of the pulp chamber along mesiodistal axis into 1 mm sections and, afterwards, into 1 × 1 × 1 mm^3^ enamel and dentine cubes under permanent water cooling. Dehydration was performed stepwise by immersing the compartments in graded solutions of ethanol (60 vol%, 4 h; 80 vol%, 12 h; 100 vol%, 12 h) and acetone (100 vol%, 12 h) at 4 °C.

### 2.2. Enamel and Dentine Ultrathin Section Preparation by Ultramicrotomy

The resulting dehydrated tissue blocks of enamel and dentine compartments were embedded in thermally cured Epon 812 (Serva, Heidelberg, Germany). Subsequent to their infiltration in a mixture of Epon 812 and 100% acetone (50:50 by volume) for 12 h at room temperature, the enamel and dentine compartments were infiltrated in Epon 812 for 6 h. Specimens were transferred to a silicon mold, appropriately oriented using a stereo zoom microscope and covered with Epon 812. The specimens were cured in an oven at 60 °C for 48 h. The resin initially became more viscous at the polymerization temperature and allowed for a marginal, leakage-free bond between human dental tissue and epoxy resin. To expose the area of interest, the resin-embedded dental tissue blocks were polished with silicon carbide abrasive paper (TegraPol-15, TegraForce-1, P4000; Struers GmbH, Willich, Germany) under constant water irrigation and, finally, polished with Topol 1 (Buehler ITW Test and Measurement GmbH, Düsseldorf, Germany); blocks were manually trimmed with a razor blade (American Safety Razor Company, Verona, WI, USA) to trim away excess resin. Ultrathin TEM sections (70–90 nm) of non-decalcified human enamel and dentine were prepared using a diamond knife (Diatome, Biel, Schweiz) on an ultramicrotome (Reichert Ultracut S; Leica, Wetzlar, Germany) at a cutting speed of 0.7 mms^−1^ and were floated on distilled water (pH 7.0). TEM sections were transferred to formvar-coated 200 mesh copper grids using a Perfect Loop (Diatome) and were finally air-dried on filter paper.

### 2.3. Enamel and Dentine Ultrathin Section Preparation by In Situ FIB Lift-Out Technique

In Situ FIB lift-out specimen preparation was applied using a dual-beam system (Quanta 3D 200i; FEI, Eindhoven, The Netherlands) equipped with a focused gallium ion beam source and a platinum gas injection system to produce high-quality TEM lamellas (approximately 90-nm thin) showing the ultrastructural morphology in detail.

Taking the tissue blocks, which were previously processed for ultrathin sections, excess resin was removed from one side of the block, ensuring exposure of the area of interest. The opposite side of the block was glued to the surface of a 12.5 mm aluminum SEM specimen stub using a carbon-adhesive disc (Leit-Tab; Plano, Wetzlar, Germany) and conductive copper tape (Plano).

To avoid charging and beam drifting during the FIB preparation, the specimens were covered with a gold layer using a sputter coater (Cressington Sputter Coater 108auto; Cressington, Watford, UK). Coarse milling was carried out using a voltage of 30 kV. The ion beam current was set to 3.0 nA and reduced stepwise to 1.0 nA. For the final polishing, the ion beam current was set to 0.5 nA and reduced stepwise to 0.03 nA. FIB milling was documented in situ by a scanning electron microscope equipped with an ETD (SE) detector used in high-vacuum mode. High-resolution imaging during coarse milling was carried out at a beam voltage of 10 kV and a spot size of 4.0 (0.23 nA), whereas final polishing was performed at a beam voltage of 5 kV and a spot size of 3.0 (41 pA).

The thickness of the FIB lamellas was measured using a scanning electron microscope (Quanta FEG250; FEI, Frankfurt am Main, Germany) at a 10 kV accelerating voltage and a spot size of 3.0.

### 2.4. TEM Imaging and Analysis

For high-resolution imaging, ultramicrotome-prepared ultrathin sections and FIB-prepared lamellas were examined by transmission electron microscopy (EM 902 and LIBRA 120; Carl Zeiss Microscopy, Jena, Germany) operated at 80 and 120 kV, respectively.

### 2.5. Microscopic Examination

To compare the sections prepared by ultramicrotomy and the TEM lamellas prepared by in situ FIB lift-out, as well as to demonstrate specific preparation artefacts, a further microscopic examination was performed. The data used to evaluate TEM sample preparation were obtained by scanning electron microscopy (Quanta FEG250; FEI, Frankfurt am Main, Germany) at a primary beam energy ranging from 5 to 10 keV with a large field detector (LFD). Surface morphology was examined using reflected-light microscopy (LEXT OLS 3000; Olympus, Hamburg, Germany), confocal laser scanning microscopy (LEXT OLS 3000; Olympus, Hamburg, Germany) and atomic force microscopy (Nanowizard I^®^, JPK-Instruments, Berlin, Germany). High-resolution AFM images were generated in contact mode using a standard CSC37 cantilever (Ultrasharp CSC37/ no AL, MicroMasch, Tallin, Estonia) with the following parameter settings: cut-off frequency, 150 Hz; amplification factor, 0.05; scan size from 100 µm × 100 µm to 2 µm × 2 µm; scan rate, 0.5 Hz; setpoint, 0.5 V at Vsum = 1.5 V; resolution, 512 × 512. AFM measurements were performed using JPK data-processing software (v.5.0.97).

## 3. Results

### 3.1. TEM Preparation Protocol of Dental Mineralized Enamel and Dentine

We developed a protocol for TEM sample preparation of human mineralized dental enamel and dentine. The complete workflow for specimen processing, sectioning and FIB/TEM lamella production, as well as the evaluation of the sample preparation, is introduced in a graphical overview in Figure 2.

### 3.2. Dental Enamel Ultrathin Sections Prepared by Ultramicrotomy: TEM

The transmission electron micrographs of sections and lamellas prepared by ultramicrotomy and FIB are compared in Figure 3 for mineralized human enamel. The results from sections prepared by ultramicrotomy are shown in Figure 3a,c,e. Enamel electron sections with transparent fragments had a limited size (approximately 2 µm of height and varying length across the resin/enamel interfaces), whereas larger areas were only rarely obtained (see Figure 7c). Numerous hydroxyapatite crystals forming nanofibrils with an aligned course were obvious (Figure 3a,c). When observed in the TEM, the enamel hydroxyapatite crystals, which are approximately oriented perpendicular to the enamel surface, show a size gradient towards the outer enamel surface (Figure 3a). The thickness of enamel hydroxyapatite crystals increases from the dentine–enamel junction (DEJ) towards the tooth’s surface. The high magnification of these hydroxyapatite crystals (Figure 3e) revealed that single hydroxyapatite crystals were covered with nanoparticles between 10 and 30 nm in size. Our results show that ultramicrotomy produced changes such as delamination of hydroxyapatite crystals and the organic matrix in human mineralized enamel (Figure 3a).

### 3.3. Dental enamel Ultrathin Sections Prepared by Focused Ion Beam: TEM

In contrast, the in situ FIB lift-out-prepared enamel (Figure 3b,d,f) obtained from the same tissue block was fully intact, showing few changes, such as crystal fracturing and delamination, in the micro- and nanostructure. The FIB–TEM lamellas can be thinned, positioned and adapted for TEM imaging. Enamel–FIB–TEM lamella thickness ranged from 175 nm to 242 nm. The dimensions of the electron transparent area were approximately 5 µm × 8 µm. As can be seen from Figure 3b, the hydroxyapatite crystals show a different alignment in the area of the prisms (PS) and the interprismatic substance (IPS), respectively. Distinct from the ultramicrotome-prepared enamel, in situ FIB lift-out-prepared enamel revealed no obvious nanoparticles on the surface of the hydroxyapatite crystals (Figure 3f). A high level of magnification revealed that hydroxyapatite crystals formed nanofibrils with an aligned course (Figure 3d), as well as single-electron transparent gaps between hydroxyapatite crystals, similar to the findings in the ultramicrotome-prepared enamel sections.

### 3.4. Dental Dentine Ultrathin Sections Prepared by Ultramicrotomy: TEM

The transmission electron micrographs of the sections and lamellas prepared by ultramicrotome and FIB are compared in Figure 4 for mineralized human dentine. Using ultramicrotomy, it was possible to prepare large-electron transparent dentine sections measuring approximately 400 µm × 300 µm. The micrographs indicate that the peritubular dentine consists of tightly packed hydroxyapatite crystals (electron-dense area indicated by dotted line; Figure 4a).

### 3.5. Dental Dentine Ultrathin Lamella Prepared by Focused Ion Beam: TEM

The thickness of dentine FIB–TEM lamellas varied between 80 nm and 134 nm. The dimensions of the electron transparent area were similar to those of enamel FIB/TEM lamellas, at approximately 5 µm × 8 µm. In contrast to ultramicrotomy-prepared sections of dentine (Figure 4c), in situ FIB lift-out-prepared intertubular dentine (Figure 4d) reveals an ultrastructure with a more reticular configuration. At the nanoscale, this structure shows an anisotropic alignment of the needle-like hydroxylapatite crystals, which are embedded in a three-dimensional network of collagen fibrils (arrows, inset in Figure 4c). Comparing ultramicrotome-prepared dentine sections (Figure 4a) and in situ FIB lift-out-prepared dentine (Figure 4b) at the peritubular and intertubular dentine interface, the higher mineralization of the peritubular dentine is clearly visible when using both preparation methods in TEM.

Table 1 presents a summary of the comparison results between ultramicrotomy and focused ion beam preparation regarding human mineralized enamel and dentine.

### 3.6. Dental Enamel Block Surface Morphology after Mechanical Polishing and Ultramicrotomy: Light Microscopy, CLSM, AFM

The block surfaces of Epon-embedded dental tissue compartments were examined by light microscopy after mechanical polishing following ultramicrotomy. In addition, the surface roughness was monitored by atomic force microscopy (AFM) (Figure 5).

Mechanical polishing caused scratches and polishing marks on the enamel surface and obscured detailed microscopic observations of the structural configuration of the dental biomaterial, e.g., of an enamel specimen (Figure 5a–c). In contrast, detailed structural information was obtained from the smoothed specimen surface after preparation in the ultramicrotome with light microscopy, CLSM and AFM (Figure 5d–f). Numerous prisms and an interprismatic substances could be differentiated (Figure 5d–f). The horizontal height profile of the mechanically polished enamel surface and the same enamel sample after ultramicrotome preparation, as measured by AFM, showed the surface roughness (Ra) at the microscale in detail (Figure 5g,h). The exemplary determined maximum height difference of 1.02 µm for mechanically polished enamel suggested that during ultramicrotomy, the approximately 10–15 sections have to be discarded. After preparation by ultramicrotomy, the enamel exhibited a relatively smooth surface, with the maximum roughness varying between 8.3 and 9.9 nm. Taken together, when observed by a combination of various microscopes, the structural configuration of enamel could be investigated at the microscale.

### 3.7. Dental Dentine Block Surface Morphology after Mechanical Polishing and Ultramicrotomy: Light Microscopy, CLSM, AFM

Figure 6 depicts the dentine block surface morphology after mechanical polishing and ultramicrotomy. Analogous to the enamel block surface morphology, the mechanically polished dentine was shown to cause scratches and polishing marks on the dentine surface (Figure 6a–c). In contrast with enamel, it is possible to see dentine’s microstructure on mechanically polished block surfaces. Numerous dentinal tubules are obvious. After ultramicrotomy, the dentine surface is smoothened, without scratches and polishing marks (Figure 6d–f).

### 3.8. Ultramicrotome-Induced Artefacts: TEM, SEM

Characteristic ultramicrotome-induced artefacts of enamel and dentine are illustrated in Figure 7. Scanning electron microscopy (Figure 7a) of ultramicrotome-prepared enamel revealed that the specimen was only preserved at the enamel–epoxy–resin interface, indicating that TEM examination would only be possible at the edges of the specimen with optimal resin penetration. The enamel in the center of the ultrathin section became lost during the preparation process when the sections were floating in the water in the trough of the diamond knife. Ultramicrotomy thus provided limited areas of enamel, hampering insights into its ultrastructure over larger distances. When observed with TEM, an alternating electron transparency was clearly visible in ultramicrotome-prepared enamel sections, indicating so-called chatter marks (Figure 7c). With dentine specimens, the sections covered the entire specimen over a larger area, indicating good resin penetration. Our results show that ultramicrotomy-induced artefacts, such as section folding (Figure 7b) and local cracks, e.g., within the peritubular dentine (Figure 7d), still exist in dentine samples.

### 3.9. Focused Ion-Beam-Induced Artefacts: TEM, SEM

The FIB method (Figure 8a,b) allows for a precisely localized preparation with few mechanical but some thermal artefacts, which cause TEM lamella bending (Figure 8b,c) and, consequentially, a shorter focus depth at these lateral image margins (Figure 8d). As shown in Figure 8e, FIB-prepared enamel sections display small voids that can be observed in hydroxyapatite crystals, particularly at the margin of the electron-transparent area. When inspected for a longer period of time by TEM, these voids may be conflated into larger holes (Figure 8f), destroying the ultrastructure of the dental hard tissue. Therefore, both ion beam and electron beam may induce artefacts into the TEM lamella.

## 4. Discussion

This study is novel regarding the systematic and detailed comparison of TEM sections of human mature mineralized enamel and dentine prepared by the in situ FIB lift-out technique and ultramicrotomy using a combination of various microscopic examination methods, including light microscopy, CLSM, AFM, SEM and TEM. Comparative studies using various microscopic methods with respect to the analysis of human mineralized enamel and dentine are largely lacking to date.

Another major objective of the present study was to evaluate the published TEM preparation protocols of human mineralized dental tissues (enamel, dentine) and to derive a protocol to prepare mineralized enamel and dentine for electron-microscopic visualization to obtain the highest possible native structure information. Several groups have published heterogeneous protocols for preparing dental tissues for transmission electron microscopy [16,17,18,19,20,21,22,23]. A major purpose of our research was to deduce the most favorable preparation conditions for TEM preparation and the ultrastructural imaging of mineralized human dental tissues.

Prior to electron microscopic examination, dental tissues must be chemically fixated to preserve the organic matrix [16,20,23,26,27]. Glutardialdehyde (2.5% by volume) is a suitable fixative for electron microscopy examinations in dental research [28]. A commonly used fixative in dental research is modified Karnovsky’s fixative [7,23]. However, both fixatives vary in osmolarity, and studies examining the effect of different fixatives on dental ultrastructure and mechanical parameters do not yet exist.

The embedding process affects the ultramicrotomy results. Several embedding mixtures are used in dental in vitro studies [29,30,31]. To ensure high quality and ultrathin sections, embedding methods in the literature were reviewed and tested. Our results showed that thermally cured Epon 812 is the embedding medium of choice for dental tissues due to its sufficient bond to enamel and dentine without clefts. The existence of this effect implies that thermally cured Epon 812 shows a lower polymerization shrinkage. The data obtained in this study indicate that epoxy resin does not infiltrate mineralized dental enamel during the embedding procedure, whereas dentine shows a good infiltration behavior. These findings are essential, as they imply the importance of a cleft-free bond between enamel and the epoxy resin when TEM preparation of enamel is performed using ultramicrotomy. For this purpose, it is advantageous to examine the interface between dental tissue and epoxy resin using microscopic methods, e.g., light microscopy, CLSM and AFM, as described above.

TEM is a high-resolution microscopy technique used to examine the ultrastructural morphology of materials at the nanoscale [20,31,32,33]. In dental research, TEM is the microscopy method of choice to investigate the ultrastructural configuration of hard tissues (enamel, dentine) [23,33,34,35], as well as their interfaces with biomaterials [28,36,37]. Ultramicrotomy using a diamond knife is the conventional means of obtaining ultrathin sections (approximately 70–90 nm) for TEM, with numerous studies on enamel and dentine, as well as their interfaces, reported in the literature [13,21,22,29,38]. The TEM preparation of mineralized human enamel and dentine using ultramicrotomy is an extremely challenging process. In the literature, the ultramicrotomy of dental hard tissues is reported to affect TEM preparation and frequently introduces artefacts, such as chatter marks and mechanical deformation [22,28,36]. Our observations are in agreement with previous reports confirming such specific artefacts, as shown in Figure 7.

The FIB technique, as well as the broad ion beam (BIB) technique, is an alternative TEM preparation method to conventional ultramicrotomy. Concerning the ultrastructural morphology of tooth–biomaterial interfaces, there seem to be no major differences between FIB- and BIB-prepared ultrathin sections. Coutinho et al. [28] reported that the main structures imaged in ultramicrotome-prepared TEM sections could also be detected on FIB/BIB-based structures. At present, FIB preparation is easily available to researchers worldwide. The FIB preparation costs are dependent on the FIB system and the experience of the human operator. In our experience, FIB preparation is 10 times more expensive than preparation using ultramicrotomy.

In contrast to the published studies, we compared the TEM preparation methods using the same specimen blocks as those used for ultramicrotomy and the in situ FIB lift-out technique. In general, the sections prepared using FIB have a smaller field of view (approximately 5 µm × 8 µm) compared to ultramicrotomy (several 100 µm). Regarding human mature mineralized dentine, our results confirmed this general assumption. Concerning mineralized human enamel, the in situ FIB lift-out sections had a larger field of electron transparency (5 × 8 µm) compared to most enameled sections, which were prepared by ultramicrotomy; these rarely reached larger electron-transparent areas (Figure 7 c) and were frequently limited to much smaller sizes of approximately 2 µm height and varying lengths across the resin/enamel interfaces. Furthermore, in agreement with the literature [25], FIB–TEM lamellas were thicker (dentine: between 80 nm and 134 nm; enamel: from 175 nm to 242 nm, as measured by SEM) than ultramicrotome-prepared sections, which ranged from 70 to 90 nm when estimated from the interference color of dentine sections on the water surface. The thickness range of FIB–TEM lamellas was determined by local variations in microhardness within a tooth, which causes diverse milling depths of the focused ion beam. As a consequence of TEM lamella bending during ion beam milling, different local milling depths were observed. The local thickness range within one FIB–TEM lamella was obvious in SEM micrographs and can be greatly reduced by using sufficiently low milling doses. Our results are in line with the results of Jantou et al. [25] and confirm that the FIB-prepared TEM sections are grossly more intact compared to sections prepared using ultramicrotomy. In this study, the FIB preparation technique (FIB/SEM) exposed major differences for human non-decalcified enamel and dentine concerning the ultrastructural information compared to ultramicrotome-prepared sections.

The TEM preparation challenge is compounded by the fact that mineralized enamel is the hardest tissue in the human body. The published data [28] indicate a diamond-knife-induced fracturing mechanism during sectioning. To confirm this assumption and investigate this fracturing mechanism in more detail, we examined the block surface morphology of enamel subsequent to mechanical polishing (Figure 5a,b,c) and after ultramicrotomy (Figure 5d,e,f). After the ultramicrotomy-based preparation, the maximum roughness varied between 8.3 and 9.9 nm as a consequence of the fracturing mechanism of enamel hydroxyapatite crystals. Hydroxyapatite crystallites of the prismatic substance (darker area), oriented orthogonally towards the surface, were fractured approximately 10 nm deeper in comparison to the crystallites of the interprismatic substance (brighter area), which were aligned nearly parallel to the specimen surface. This suggests that the hydroxyapatite crystallites of the interprismatic substance are cleaved during preparation and are located higher after preparation. However, the directionality of the hydroxyapatite crystals leads to different cutting rates at different material orientations, leading to artefacts being induced by ultramicrotomy. Our results show that ultramicrotome-prepared enamel blocks had an approximately 50× smoother surface compared to the mechanically polished surfaces. A further advantage is that structural features (prisms, interprisms) were very microscopically visible on ultramicrotome-prepared surfaces. In general, focused ion beam preparation can be applied to mechanically polished biological tissues. For site-specific TEM preparation, previous knowledge of the position of the region of interest in the specimen is essential. Hence, we recommend the use of pre-ultramicrotomy as a polishing technique in future FIB applications to improve the localization of structural target features.

Previous reports cover ivory dentine and dehydrated maxillary/mandibular human premolars, using TEM [22,25] in combination with X-ray energy-dispersive spectroscopy (XEDS) and electron energy-loss spectroscopy (EELS) [22] to evaluate the preparation results. Both studies used ultramicrotomy and FIB preparation to compare dehydrated human mineralized enamel and dentine [22,25]. Ultramicrotomy-prepared enamel, when analyzed by TEM, showed a similar nanostructure to that reported by Srot et al. [22]. The enamel sections prepared by ultramicrotomy showed artefacts such as delamination of the hydroxyapatite crystals and the organic matrix in human mineralized enamel. The artefacts induced by the cutting process using a diamond knife can also be supplemented by transfer artefacts.

An important implication of our findings is that the enamel sections prepared by ultramicrotomy revealed hydroxyapatite crystals that were covered with numerous nanoparticles with sizes of 10 and 30 nm, which are reminiscent of mineralization and growth centers, similar to those generated in an in vitro study by Deshpande et al. (2010) [39]. More investigations on deciduous and permanent teeth of different ages will be needed to verify this assumption. There was an evident relationship between the size of enamel hydroxyapatite crystals and localization within the enamel. The length and thickness of enamel hydroxyapatite crystals increased from the DEJ towards the outer enamel surface.

Concerning FIB preparation, our investigations revealed similar artifacts as those found in a study by Srot et al. [22]. So-called artificial FIB-induced voids (up to 10 nm) in enamel hydroxyapatite crystals were also detected in our study, at a considerably higher resolution. At the margin of the electron-transparent area, numerous voids were detected due to the fact that the focused ion beam stayed in these areas for longer during the milling process. These artefacts constitute morphological damage inside the hydroxyapatite crystals. However, their frequency of occurrence could be minimized by a slower milling rate, using a low-ion beam current. In contrast to ultramicrotomy, the hydroxyapatite crystals of in situ FIB lift-out-prepared enamel showed no nanoparticles on the surface.

In agreement with the previously published literature, the SEM micrographs of our ultrathin dentine sections indicated the characteristic artefacts induced by ultramicrotomy-like cutting marks and section folding or local microcracks in the peritubular dentine [22]. At the nanoscale, our TEM examination confirmed the results of Jantou et al. [25], showing some structural damage in ultramicrotome-prepared TEM sections. The anisotropic alignment of the needle-like hydroxyapatite crystals, which are embedded in a three-dimensional network of collagen fibrils, is only obvious in FIB–TEM lamellas and is comparable to previous reports on dehydrated human dentine and ivory dentine [22,25].

Mature human mineralized enamel and dentine have a unique structure. The anisotropy, which is manifested at all levels from the nanoscale, up to the macroscale, conscientiously reflects its biomechanical properties. At present, it is not possible to entirely exclude ultrastructural changes in the dental hard tissue due to storage and preparation limitations. Hence, the term “native” should be used with care. Artificial saliva mixed with glutardialdehyde (2.5% by volume) is a very suitable near-native fixative for human mineralized tissue. A critical discussion of potential artefacts and a comparison of different TEM preparation techniques permits a successful approximation of the native structure of dental biomaterials. In oral and maxillofacial surgery, implant rehabilitation and bone grafting is a standard treatment [40]. Here, the FIB–TEM method is useful to investigate the implant–bone interface in cases of dental implant failure. In particular, the osseointegration of the failed implant can be analyzed by FIB–TEM [41]. Moreover, FIB can be useful for the visualization of biological soft tissues. Bushby et al. (2011) [42] described an FIB/SEM technique for 3D imaging of a chicken cornea.

## 5. Conclusions

Comparison of the described TEM preparation methods shows that ultramicrotomy can be used for human mineralized dentine cutting, with the possibility of a few mechanical artefacts; however, this is limited to human mineralized enamel. The FIB method shows strong potential for preparing both mineralized biomaterials for TEM imaging and analysis. In conclusion, our study shows that, unfortunately, no single preparation technique is capable of optimally preserving each of the structural features of dental tissues. Only by using several different preparation methods and exploiting appropriate preparation conditions does it seem possible to assemble a comprehensive overview of the structure of dental tissue in a near-native state. Taken together, our study presents an optimized procedure for the structural analysis of mineralized enamel and dentine from human teeth.

## Figures and Tables

**Figure 1 materials-15-03084-f001:**
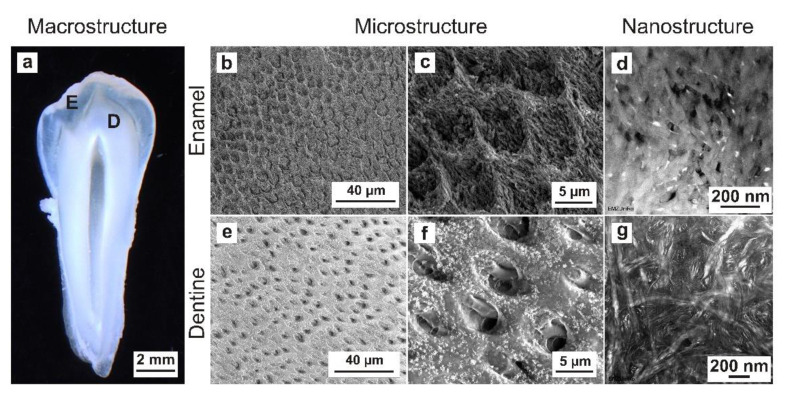
The structural hierarchy of mature human enamel and dentine from the macroscale to nanoscale. (**a**) Stereo-zoom microscopy of a tooth slice from a premolar; thickness, 1 mm; view from mesial, (E) enamel and (D) dentine. (**b**) SEM micrograph of enamel. The alternating orientation of the prism rods (level 6) reveals the Hunter–Schreger bands (level 7). (**c**) SEM micrograph of enamel showing interprisms and prisms (level 5) formed by hydroxyapatite crystallite nanofibrils (level 2–4). (**d**) TEM micrograph of enamel showing hydroxyapatite crystallites (level 1). (**e**) SEM micrograph showing numerous dentinal tubules, peritubular dentine and intertubular dentine. (**f**) SEM micrograph illustrating dentinal tubules and odontoblast processes. (**g**) TEM micrograph of the nanostructure of intertubular dentine.

**Figure 2 materials-15-03084-f002:**
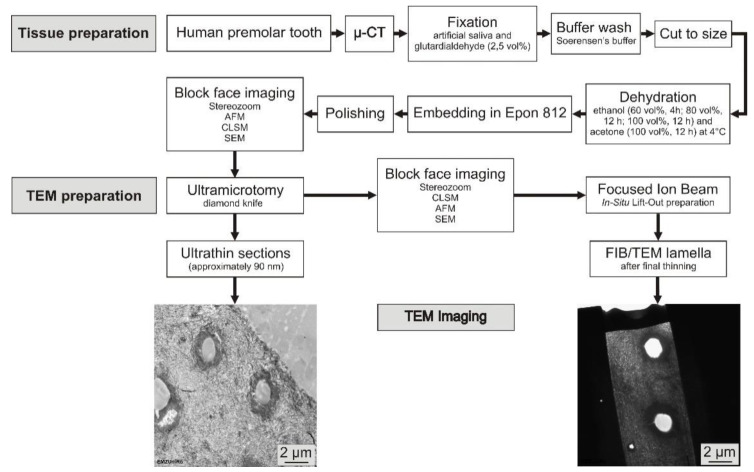
Experimental workflow of human mineralized enamel and dentine preparation using ultramicrotomy and in situ FIB lift-out technique for TEM imaging. For inspection and quality control in tissue preparation, multimicroscopic imaging was used (Stereozoom, AFM, CLSM, SEM).

**Figure 3 materials-15-03084-f003:**
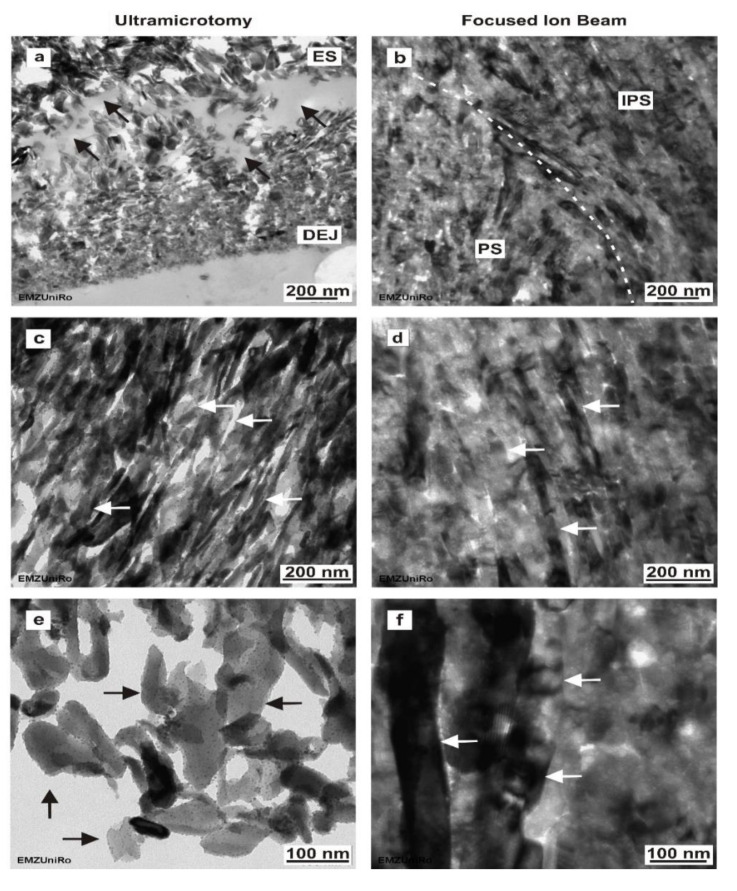
Comparison of ultramicrotome and in situ FIB lift-out-prepared human mineralized enamel. (**a**) Ultramicrotome-prepared enamel section depicting the increasing size gradient of enamel hydroxyapatite crystals towards the outer enamel surface (ES). Delamination of the hydroxyapatite crystals and the organic matrix, indicated by arrow heads. (**b**) In situ FIB lift-out-prepared enamel section at the interface (white broken line) between an enamel prism (PS) and the interprismatic substance (IPS), showing the different orientation of hydroxyapatite crystals in PS and IPS. (**c**) Ultramicrotome-prepared enamel section showing hydroxyapatite crystals forming nanofibrils with an aligned course (arrows). (**d**) In situ FIB lift-out-prepared enamel section showing hydroxyapatite crystals (arrows) inside the interprismatic substance. (**e**) High magnification of an ultramicrotome-prepared enamel section reveals single hydroxyapatite crystals (arrows) that are covered with nanoparticles between 10 and 30 nm in size. (**f**) High magnification of an in situ FIB lift-out-prepared section showing the enamel nanostructure in detail, with no nanoparticles on the surface of the hydroxyapatite crystals (arrows).

**Figure 4 materials-15-03084-f004:**
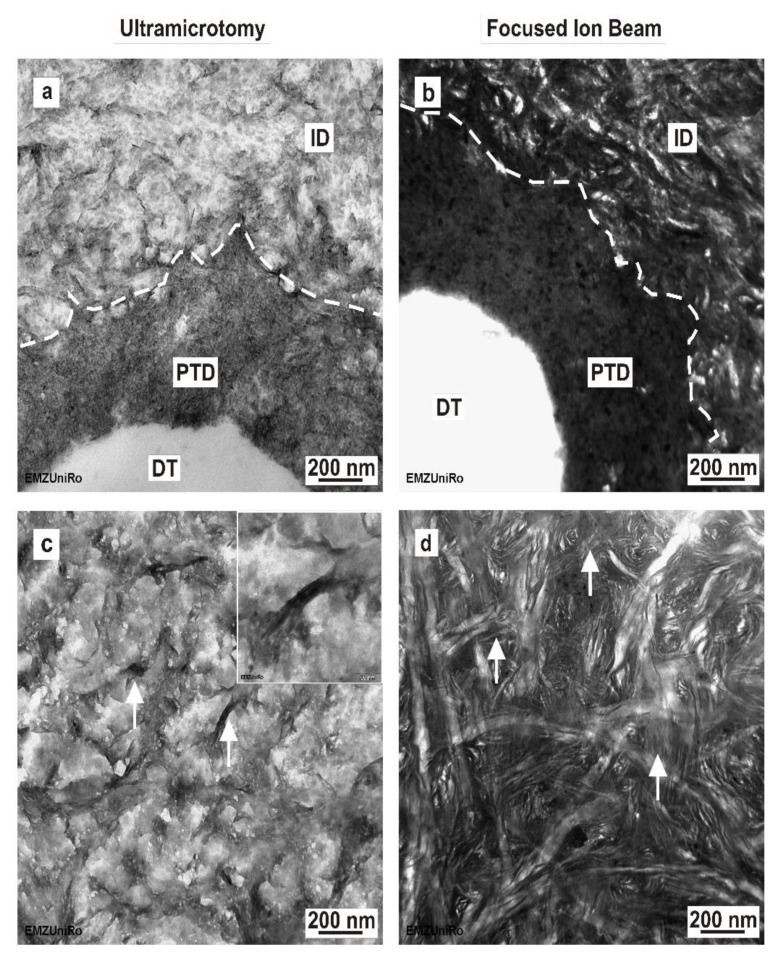
Comparison of ultramicrotome and in situ FIB lift-out-prepared human mineralized dentine. (**a**) Ultramicrotome-prepared dentine section showing the peritubular (PTD) and intertubular dentine (ID) interface; (DT) dentinal tubule (**b**) In situ FIB lift-out-prepared dentine section showing the highly mineralized peritubular dentine (PTD) and the less mineralized intertubular dentine (ID); (DT) dentinal tubule. (**c**) Ultramicrotome-prepared intertubular dentine. The inset in c is a magnified TEM image of a hydroxyapatite crystal bundle. (**d**) In situ FIB lift-out-prepared intertubular dentine section. Arrows indicate hydroxyapatite crystal bundles.

**Figure 5 materials-15-03084-f005:**
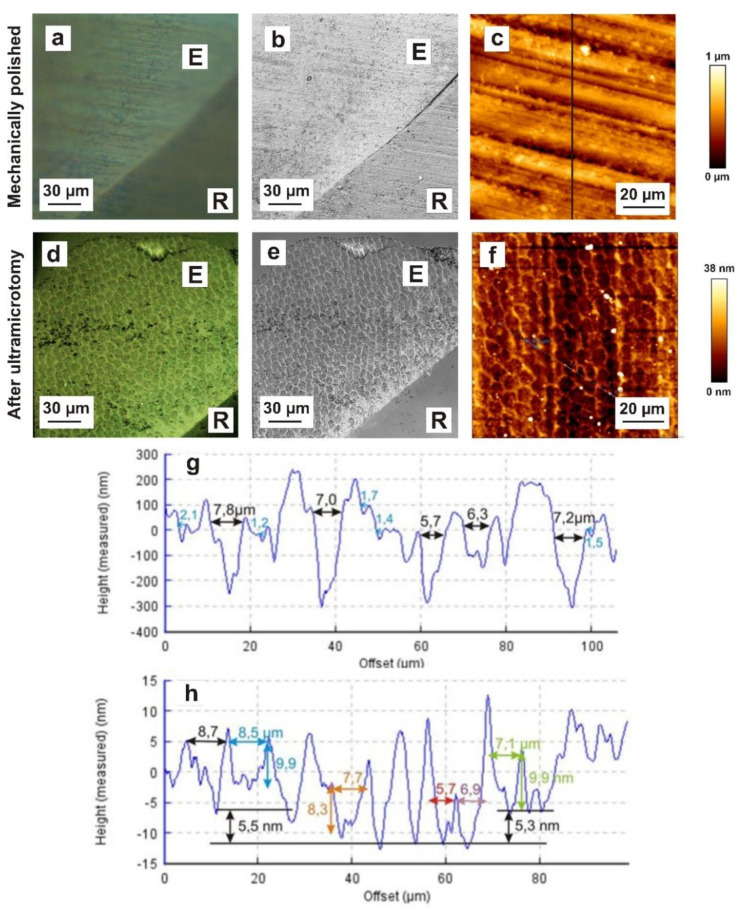
Comparison of enamel block surface morphology after mechanical polishing and ultramicrotomy. (**a**) Reflected-light microscopy (differential interference contrast mode) of mechanically polished enamel showing polishing marks on the enamel surface with enamel (E) and resin (R). (**b**) Confocal laser scanning microscopy of mechanically polished enamel showing scratches and polishing marks. The microstructure of enamel (prisms and interprismatic substance) is not obvious with enamel (E) and resin (R). (**c**) AFM micrograph of mechanically polished enamel depicting scratches and polishing marks. (**d**) Reflected-light microscopy of the enamel block surface after ultramicrotomy, showing the microstructure of enamel (prisms and interprismatic substance) with enamel (E) and resin (R). (**e**) Confocal laser scanning microscopy of the enamel surface after ultramicrotome preparation, depicting numerous prisms and the interprismatic substance with enamel (E) and resin (R). (**f**) AFM micrograph of the enamel surface after ultramicrotome preparation showing numerous prisms (dark red area) and the interprismatic substance (yellow area). (**g**) Horizontal height profile of the mechanically polished enamel surface measured by AFM, showing the surface roughness at the microscale in detail. (**h**) Horizontal height profile of the same enamel sample after ultramicrotome preparation, measured by AFM.

**Figure 6 materials-15-03084-f006:**
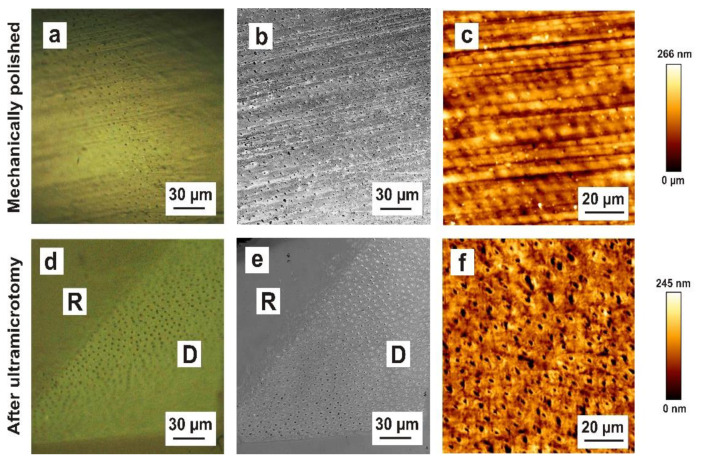
Comparison of dentine block surface morphology after mechanical polishing and ultramicrotomy. (**a**) Reflected-light microscopy (differential interference contrast mode) of mechanically polished dentine showing polishing marks on the dentine surface. (**b**) Confocal laser scanning microscopy of mechanically polished dentine showing scratches and polishing marks. The microstructure of dentine (dentinal tubules and intertubular substance) is obvious. (**c**) AFM micrograph of mechanically polished dentine depicting scratches and polishing marks. (**d**) Reflected-light microscopy of the dentine block surface after ultramicrotomy, showing the microstructure of dentine (dentinal tubules and intertubular substance) with dentine (D) and resin (R). (**e**) Confocal laser scanning microscopy of the dentine surface after ultramicrotome preparation depicting numerous dentinal tubules and the intertubular substance with dentine (D) and resin (R). (**f**) AFM micrograph of the dentine surface after ultramicrotome preparation showing numerous dentinal tubules and the intertubular substance.

**Figure 7 materials-15-03084-f007:**
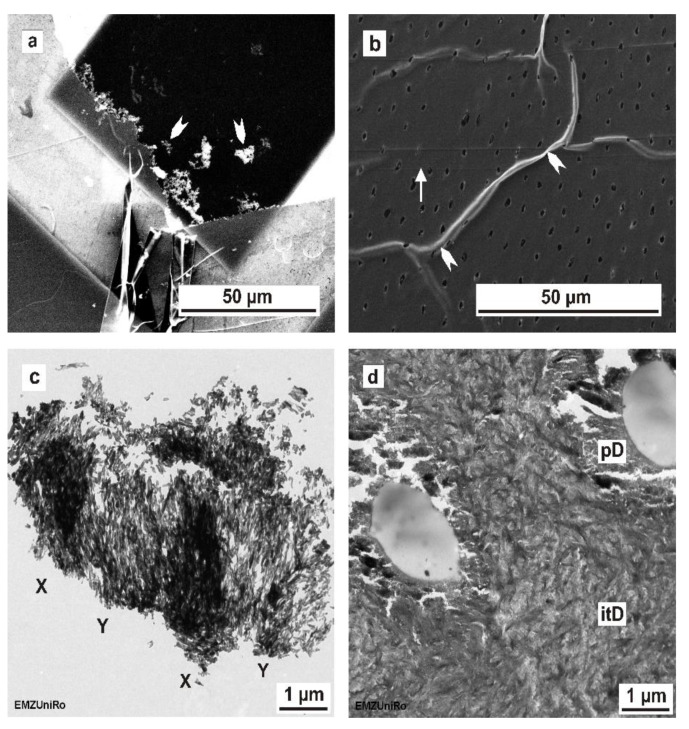
Dentine and enamel ultrathin sections showing ultramicrotome-induced artefacts. (**a**) SEM image of ultramicrotome-prepared enamel. TEM examination is only possible near the enamel–epoxy–resin interface (white arrow heads). Enamel in the center of the ultrathin section became lost during the preparation process using ultramicrotomy. (**b**) SEM image of ultramicrotome-prepared dentine showing local folding (white arrow heads) and cutting marks (white arrow). (**c**) TEM image of ultramicrotome-prepared enamel. The X–Y bending indicates chatter marks. (**d**) TEM image of ultramicrotome-prepared dentine showing local cracks at the peritubular dentine (pD), with intertubular dentine (itD).

**Figure 8 materials-15-03084-f008:**
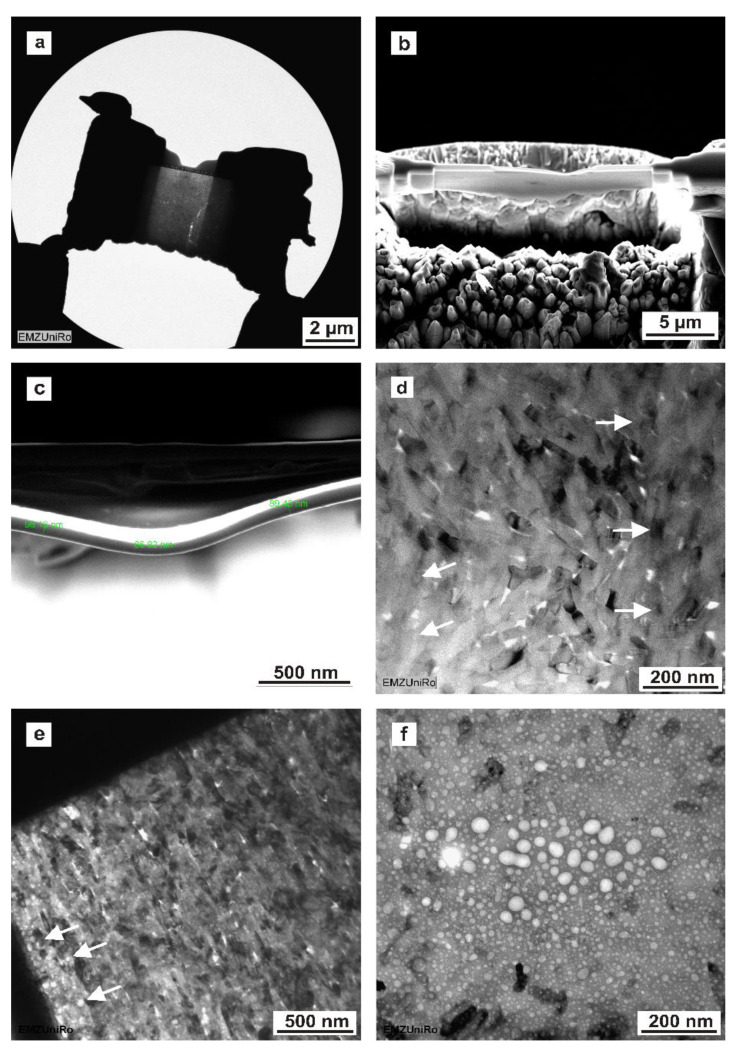
Enamel and dentine sections depicting FIB and electron-beam-induced artefacts. (**a**) TEM image of an FIB-prepared enamel lamella. (**b**) SEM micrograph of an FIB-prepared dentine section depicting FIB/TEM–lamella bending. (**c**) SEM micrograph of FIB-prepared dentine showing enlarged FIB/TEM lamella bending. (**d**) TEM image of FIB-prepared enamel. The arrow heads indicate a short focus depth at the lateral image margin due to the FIB/TEM lamella bending. (**e**) TEM image of FIB-prepared enamel depicting the numerous voids introduced during FIB preparation in enamel hydroxyapatite crystals. These voids are more clearly observed at the margin of the electron-transparent area (arrow heads). (**f**) TEM image of FIB-prepared enamel showing FIB/TEM lamella destruction induced by TEM inspection. Small initial voids induced during FIB milling create beam-sensitive areas that conflate into holes and destroy the hard tissue’s ultrastructure during TEM analysis.

**Table 1 materials-15-03084-t001:** Comparison of the TEM preparation methods, ultramicrotomy and focused ion beam, on human mature mineralized enamel and dentine.

	Ultramicrotomy	Focused Ion Beam
Preparation method	Time-consuming (i.e., specimen preparation), labor-intensive approach; extremely high level of expertise in ultramicrotomy required; low-budget.	Targeted TEM preparation
Ultrastructure	Enamel: limited practicability, nanoparticles on hydroxy-apatite crystals; Dentine: partial near-native areas.	Enamel and near-native dentine
Dimensions of the transparent election areas	Enamel: large areas rarely obtained; small areas approx. 2 µm in height and have varying length across the resin/enamel interfaces. Dentine: approximately 400 µm × 300 µm	5 µm × 8 µm
Potential artefacts	Enamel: crystal fractures, chatter marks Dentine: delamination, local cracks at the peritubular dentine, local folding, cutting marks	Lamella bending → short focus depth at the lateral image margin Thermally induced artefacts, i.e., voids in crystals

## Data Availability

Not applicable.

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
