# Peer review of "Comparative Sample Preparation Using Focused Ion Beam and Ultramicrotomy of Human Dental Enamel and Dentine for Multimicroscopic Imaging at Micro- and Nanoscale"

_materials, 2022, doi:10.3390/ma15093084_

Round 1
Reviewer 1 Report
The paper is well written and I think it offers interesting concepts for the relatively little addressed field of dental microscopy.
Reviewer 2 Report
This paper describes and compares different techniques of sample preparations for TEM ultrastructural observations of enamel and dentine. It identifies advantages and disadvantages of different techniques for sample preparations and for observations at both the micro- and nano- scales, and provides useful information for readers interested in these techniques. However, the paper is very difficult to read in its current form and needs to be significantly improved to be publishable. The English language must be edited extensively to improve the readability. The methods, results and discussion sections should be arranged in some logical orders to allow the readers to follow.
In the Methods section, I suggest that the authors arrange the contents by tissue types and preparation methods. For example: Dental Enamel - Ultrathin enamel section preparations by ultramicrotomy - Enamel preparations by in-situ FIB lift-out technique; Dentine - Ultrathin dentine section preparations by ultramicrotomy - Dentine preparations by in-situ FIB lift-out technique.
In the Results section, I suggest that the authors arrange the findings in the same order as in the Methods section and describe the results of TEM (AFM, CLSM, SEM) observations under each preparation technique. For example: Dental Enamel - Ultrathin enamel section prepared by ultramicrotomy: TEM (CLSM, AFM, SEM) - Enamel specimens prepared by in-situ FIB lift-out technique: TEM (CLSM, AFM, SEM); Dentine - Ultrathin dentine section preparations by ultramicrotomy: TEM (CLSM, AFM, SEM) - Dentine preparations by in-situ FIB lift-out technique: TEM (CLSM, AFM, SEM). It should be helpful if the images prepared by different techniques are arranged side-by-side at micro- and nano-scale levels.
As the stated purpose was to come up with the best protocol for sample preparations for TEM observation of enamel and dentine, I would suggest that the suggested protocols be clear listed in the Results section. The discussion section should focus on the advantages of these protocols.
Reviewer 3 Report
The Authors compare here the TEM-sections of mineralized human enamel and dentine prepared by focused ion beam (In-Situ Lift-Out) technique and ultramicrotomy through a combination of microscopic examination methods (SEM and TEM). Material and methods section was fully described (with the exact characteristics of the equipments. The results section and discussion were clear written. This manuscript contain new information, what can be a support of knowledge for preparing mineralized biomaterials for transmission electron microscopy imaging and analysis.
I suggest minor revision of this manuscript:
- The correction of the title is required - please check the formatting of pdf file.
- The scale bars of the figures (Fig. 1., Fig. 2) are too small. Additionally if possible please enlarge of figures 1, 2, 3 within the manuscript.
Reviewer 4 Report
The manuscript entitled „Comparative sample preparation using focused ion beam and ultramicrotomy of human dental enamel and dentine for multimicroscopic imaging at micro and nanoscale“ aimed to compare TEM-sections of mineralized human enamel and dentine prepared by focused ion beam (In-Situ Lift-Out) technique and ultramicrotomy through a combination of microscopic examination methods (scanning electron microscopy and transmission electron microscopy). The results demonstrated the conventional ultramicrotomy to be adequate method for the preparation of mineralized dentin, while there were limitations regarding enamel preparation. The FIB preparation provided TEM sections with less important structural artefacts. This paper is interesting for the part of the research community that investigates the ultrastructure of dental tissues and is well aware of the complexity of mineralized tissue preparation for TEM observation.
There are however certain concerns to be addressed:
Introduction
- Line 48, down instead of up
- Figure 1: the microscopical images could be larger for better clarity
- Line 79. A separate paragraph should start from “In comparison”, and in fact, it would be better to change the “in comparison” with “On the other hand” or similar
- Line 98. Should there be “employed” instead of “explored”?
Results
- Figure 3. How can there be a comparison between the enamel block between the mechanical polishing and ultramicrotomy if different microscopy techniques were used? For mechanical poliching light microscopy and for ultramicrotomy confocal laser scanning microscopy?
- More pointers should be added on figures 6 and 7 to better illustrate the descriptions.
- Page10, lines356-358, it seems that this sentence should be at the end of the next paragraph.
- In general, the last paragraph of the results should be revised for better clarity. Also, the first sentence of that paragraph: “Distinct from ultramicrotome prepared enamel, In-Situ FIB Lift-Out prepared enamel revealed no obvious nanoparticles on the hydroxyapatite crystals surface.” belongs to the enamel section.
Discussion
- There should be no subheadings in the discussion section.
- Lines 395-396 mention references 22 and 25, but do not discuss what these studies show and how they can be compared to the present study.
- Please group the references in the text. Example,instead of
- [20], [33], [36] and [37] in line 438, put [20,33,36,37].
- Table 1 should be placed within the results section, not the discussion section.
- There should be more information on the potential limitations of the FIB technique. How expensive is the Focused Ion Beam TEM specimen preparation? How available is it to researchers worldwide? How many samples can be obtained from one tooth section using ultramicrotomy and how many using FIB, etc…
- Lines 513-514: The authors stated that they recommend " we recommend the use of pre-ultramicrotomy as a polishing technique to future FIB applications, to improve the localization of structural target features." Does this not mean that the preparation procedure becomes even more complex?
Conclusions
- “The comparison of the described TEM prepa- 554 ration methods shows that ultramicrotomy can be used for human mineralized dentine 555 cutting with a possibility of few mechanical artefacts but is limited for human mineralized 556 The FIB method shows strong potential for preparing both mineralized bio- 557 materials for TEM imaging and analysis. In conclusion our study shows that unfortu- 558 nately no single preparation technique is capable of optimally preserving each of the 559 structural features of dental tissues. Only by using several different preparation meth- 560 ods and exploiting appropriate preparation conditions it seems possible to assemble a 561 comprehensive overview of the structure of dental tissue in a near-native state.“
“Taken together our study presented 566 an optimized procedure for the structural analysis of mineralized enamel and dentine 567 from human teeth.”
These are the only points that should be in the conclusions. Everything else can be placed as the last paragraph of the discussion.
English language should be edited throughout the manuscript.
Round 2
Reviewer 2 Report
Please make sure that the paper is truly reviewed by a professional English editor, some apparent errors remain:
Page 2, line 46: the word "restauration" means the purveying of food (as by a restaurant), it cannot possibly be correct.
Page 2, line 46, 47: This sentence is difficult to understand - I do not know what the sentence is trying to say. "The implementing" is very problematic. The noun for "implement" is "implementation", and it means to "put something into effect". "The implementing" is "the action of putting (dental materials with optimum properties) into effect" does not really connect with the complex structure of the sentence that follows, including "without prior demineralization". I do not know how to improve the sentence because I simply do not understand what it means.
I admit that I did not read the rest of paper in details as an language editor. But the paper overall has improved and I support its publication.
